# Porous Silicon Gas Sensors: The Role of the Layer Thickness and the Silicon Conductivity

**DOI:** 10.3390/s20174942

**Published:** 2020-09-01

**Authors:** Francisco Ramírez-González, Godofredo García-Salgado, Enrique Rosendo, Tomás Díaz, Fabiola Nieto-Caballero, Antonio Coyopol, Román Romano, Alberto Luna, Karim Monfil, Erick Gastellou

**Affiliations:** 1Centro de Investigación en Dispositivos Semiconductores, Benemérita Universidad Autónoma de Puebla, 14 sur y Av. San Claudio, Puebla 72570, Mexico; godofredo.gracia@correo.buap.mx (G.G.-S.); enrique.rosendo@correo.buap.mx (E.R.); tomas.diaz@correo.buap.mx (T.D.); antonio.coyopolsolis@viep.com.mx (A.C.); roman.romano@correo.buap.mx (R.R.); jose.luna@correo.buap.mx (A.L.); karim.monfil@correo.buap.mx (K.M.); 2Facultad de Ciencias Químicas, Benemérita Universidad Autónoma de Puebla, 14 sur y Av. San Claudio, Puebla 72570, Mexico; fabiola.nieto@correo.buap.mx; 3División de Tecnologías de la Información y Comunicación, Universidad Tecnológica de Puebla, Antiguo Camino a La Resurrección 1002-A, Zona Industrial, Puebla 72300, Mexico; erick_gastellou@utpuebla.edu.mx

**Keywords:** porous silicon sensor, sensor conductivity type, porous silicon layer thickness, resistive sensor

## Abstract

We studied the influences of the thickness of the porous silicon layer and the conductivity type on the porous silicon sensors response when exposed to ethanol vapor. The response was determined at room temperature (27 ${}^{\circ }$C) in darkness using a horizontal aluminum electrode pattern. The results indicated that the intensity of the response can be directly or inversely proportional to the thickness of the porous layer depending on the conductivity type of the semiconductor material. The response of the porous sensors was similar to the metal oxide sensors. The results can be used to appropriately select the conductivity of semiconductor materials and the thickness of the porous layer for the target gas.

## 1. Introduction

Gas sensing has great importance in environmental monitoring and protection. The need for cheap, small, low-power-consuming, and reliable solid-state gas sensors has grown over the years. One important research line concerns metal-oxide based sensors. The development of information technology has triggered a large amount of research worldwide to overcome metal oxide sensor drawbacks, summed up in improving the well-known ‘‘3Ss’’: sensitivity, selectivity, and stability [1]. The electrical resistance changes upon gas adsorption onto metal oxide sensors has been well studied, and detailed models regarding the sensor-gas mechanism were proposed [2,3].

The primary disadvantage is the high-power consumption as its operation is heater based. The temperature operation is approximately 200 to 400 ${}^{\circ }$C [4]. Another important research line is related to the sensing properties of porous silicon, a material obtained by chemical or electrochemical dissolution of crystalline silicon (c-Si) [5]. These works focus on development of optical or resistive devices [6,7,8]. Some of their advantages include low temperature operation, low cost, easily fabrication, and silicon technology compatibility [9,10,11,12]. The optical and electrical properties of porous silicon have been used to develop sensors with an optical or electrical response.

The pore morphology is easy to modulate during the fabrication, allowing for width and length design of the pores. The pore morphology has a direct effect on the specific area. In sensing, the specific area is the available area exposed to the target gas. The physisorption of the molecules of the target gas interact with the specific area, changing the optical or electrical properties in the porous silicon. We expected that a large specific area would drive a more intense response, and we observed that the intensity of the response depended not only on the specific area but also on the silicon conductivity type.

The surface area of porous silicon or “specific surface area”, is defined as the accessible area of solid surface per unit mass of material [13]. The specific surface area is a function of the porosity, pore size distribution, shape, size, and roughness [14]. In porous silicon, the specific surface area it is very much dependent on the method, experimental conditions, and size of the probe used [15,16,17]. In this work, porous silicon samples with a porosity of 45% from p and n conductivity types were used to fabricate sensors. The thicknesses of the porous layer varied at 1, 5, and 10 μm. Metallic electrodes were deposited on the porous silicon to study the electric response. During the gas sensing, the specific area layer was primarily affected by the physisorbed gas, and the response of the sensor was strongly dependent on the conductivity type of the semiconductor material and the nature of the target gas, as this can be an oxidizing or a reducing gas. We propose that the behavior of the response can be treated similar to the model of conduction of the metal oxide sensors [18,19,20].

## 2. Materials and Methods

### 2.1. Porous Silicon

Crystalline silicon (1 0 0) p-type (0.01–0.02 Ω·cm resistivity) and n-type (0.0015–0.004 Ω·cm resistivity) was used to fabricate the porous silicon (PS) samples. The samples were cleaned using the RCA method before the anodization [21]. The anodization conditions were settled to fabricate PS samples with 45% porosity and layer thicknesses of 1, 5, and 10 μm. The PS samples were fabricated in a Teflon cell using an electrolyte of hydrofluoric acid (HF) (EMSURE, 48%) and absolute ethanol (EtOH) (J. T. Baker, 99.98%). The PS n-type samples were fabricated using a current density of 10 mA/cm^2^ under UV illumination with an 3:1 (HF:EtOH) electrolyte proportion. The PS p-type samples were fabricated using a current density of 13.6 mA/cm^2^ in darkness with an 1:1 (HF:EtOH) electrolyte proportion. The anodization time was settled to obtain the proposed layers thicknesses, the samples features are shown in Table 1. The samples were labeled using the type of conductivity and the thickness of the porous layer. The character of the label corresponds to the conductivity type and the two digits are related to the thickness of the porous layer.

After the anodization, each PS sample was rinsed out first with EtOH to eliminate the electrolytic residues, and then with 18.2 MΩ deionized water. Finally, the porous samples were dried using an air flux. The PS was fabricated in the bulk silicon and was not separated from it. The bulk around and under the PS served as substrate, and a photograph of the PS is shown in Figure 1c. The diameter of the porous area was 18.7 mm.

### 2.2. Metallic Electrodes

A geometric pattern was designed to deposit electrodes, make electrical contact, and obtain the sensor device. The pattern was reproduced in a metallic mask, which was used during a metal deposition process. Aluminium (Sigma Aldrich (St. Louis, MO, USA), 99.999% purity) was deposited by sublimation using a JEOL JEE-420 vacuum evaporator. Each electrode consisted of two concentric no-closed rings of 1 mm width, connected by a line of the same width. Both electrodes were deposited horizontally, on the porous material. The separation between the electrodes was 1 mm, except at the end of the rings, at the corners (the separation was 1.4 mm). Figure 1a shows the design of the geometric pattern with its dimensions, and Figure 1b is a photograph of the mask used to deposit the electrodes. The light parts was where the aluminium went through to deposit, whereas the dark parts covered the porous silicon to avoid the deposit. The area of the porous silicon was 2.746 cm${}^{2}$.

The electrodes on the porous silicon samples did not touch the Si bulk, they were deposited on the porous area. Figure 1d is a photograph of a sensor with the deposited electrodes, where the diameter of the outer ring was 17 mm. After the aluminum deposit, the electrodes were annealed for one hour in a nitrogen environment at 450 ${}^{\circ }$C. A copper wire of 0.3 mm diameter was attached with Indium to each electrode on the sensors.

### 2.3. Characterization

The sensors were separately characterized into a chamber with the wires connected to terminals, which were located on one of the walls of the chamber to provide external electrical contact (see Figure 2). A BK Precision 1697 power supply was externally connected to the sensor to apply 5 V DC. A digital multimeter Protek 506 connected to a personal computer was used to measure the electrical current. The power supply, the multimeter and the sensor were connected as a series circuit.

The chamber was hermetic and was covered to make the measurements in darkness, at atmospheric pressure, and the temperature of all of the measurements was 27 ${}^{\circ }$C. A nitrogen flux was used to clean the chamber atmosphere and as a carrier gas. The nitrogen can flow directly to the chamber or through a container filled with EtOH before arrival at the chamber, carrying EtOH molecules through bubbling. During the bubbling, the EtOH was maintained at a controlled temperature into a refrigerated bath, Thermo Electron Neslab TRE7, with the aim to control the vapor pressure ($P_{EtOH}$). The container with EtOH was submerged into the bath at 25 ${}^{\circ }$C. The ($P_{EtOH}$) was calculated at 58.73 mmHg, using the Antoine equation and their respective coefficient values for ethanol [22,23]. The nitrogen flow ($F_{N}$) introduced in the reactor was kept constant at 100 cm${}^{3}$/min during the characterization, controlled with a Mathesson flowmeter. The total flow into the reactor was the sum of the flows of the carrier gas plus the EtOH vapor, in total, 108.4 cm${}^{3}$/min. The EtOH vapor flow was calculated using Equation (Equation 1)
(1)$$EtOH_{vap}=\frac{F_{N}\times P_{EtOH}}{P_{tot}-P_{EtOH}}$$
where the total pressure $P_{tot}$ was the atmospheric pressure (760 mmHg) [24]. A schematic representation of the sensing reactor is shown in Figure 2. The flow (the carrier gas plus the EtOH vapor) flowed parallel to the sensor surface, filled the reactor and left through the exhaust line.

A baseline is a reference of the electrical current through the sensor before the target gas flows into the chamber. The baseline was obtained when the electrical current was constant while nitrogen was flowing into the chamber, cleaning the surface of the sensor. The sensors were characterized in ten minute periods of exposure and recovery. In the exposure period, the sensor was in the presence of the EtOH vapor, whereas in the recovery period, the flow of the EtOH vapor was stopped, and the chamber environment was ventilated with a nitrogen flux to facilitate the desorption process and clean the environment of EtOH vapor. The switching time between the exposure and recovery period were designated as “on” and “off”, respectively. The delay from when the target gas is in the reactor until the sensor first begins to respond is the dead time.

## 3. Results and Discussion

The sensor response is the change of the electrical current with respect to the baseline, due to the EtHO vapor presence in the chamber atmosphere. In all graphs, the baseline was subtracted from the sensor response to remove any noise and compare the response of the sensors with the same conductivity type. The time at t=0 corresponded to the moment at which the EtOH vapor was introduced for first time into the chamber. The sensor recovery was the intensity of the response at the end of the recovery time compared to the intensity of the response at the start of gas exposure. The sensor fully recovered if these two intensities coincided.

The results of the n-type porous sensors are shown in Figure 3. The N10 sensor (type-n with a 10 μm porous layer thickness) presented a more intense response than the N05 and N01 sensors (type-n with 1 and 5 μm porous layer thickness). The dead time was approximately 30 s in all cases, both in the exposure and recovery periods. The response intensity increased as the thickness of the porous layer increased, showing a proportional relation.

The p-type porous sensors responses are shown in Figure 4. The response intensity of these sensors decreased as the thickness of the porous layer increased. The response was more intense in the P01 sensor (p-type with a 1 μm porous layer thickness) than in the P05 and P10 sensors (p-type with 5 and 10 μm porous layer thickness). The responses presented a longer dead time in the function of the layer thickness: the thicker the porous layer, the longer the dead time. The dead times were approximately 115, 152, and 247 s (in the first exposure period) for P01, P05, and P010, respectively. The recovery was better in P01 as the intensity of the response was closer to the intensity at the beginning of the exposure time. The dead time and recovery were inversely proportional to the thickness of the porous layer.

We proposed that the behavior of the response observed on the porous sensors could be explained using the oxide-reduction reaction, similar to the metal oxide-based gas sensors. When a p-type semiconductor material interacts with a reducing gas (EtOH), the sensing layer is depleted of charge carrier electrons (conduction electrons), resulting in a decrement of the conductivity [25,26], this is concordant with the response of the p-type porous sensors. On the other hand, the conductivity increases when an n-type semiconductor material is in presence of a reducing gas [25,26], the redox reaction increases this charge.

After the first characterization, the sensors were exposed to ethanol for an extended period to observe the stability of the response. The exposure lasted at least two hours, arbitrarily, until the sensors reached a stable response. The response of the n-type porous sensors are show in Figure 5a. The curves exhibited a stable response and a recovery without reaching the base line. The response of the p-type porous sensors are shown in Figure 5b. After two hours of exposure, the response of the p-type sensors was not stable and continued growing. The exposure was stopped after 2, 4, and 6 h for P05, P01, and P10, respectively. The three sensors recovered nearly to the base line, but only the recovery of P05 was observed, due to the long time taken for measurements. Our results showed that the stability was influenced by the conductivity type of the silicon.

Finally, the sensors with a lower response intensity, N01 and P10, for each type of conductivity, were selected to be measured in a lower ambient vapor pressure (less EtOH carried into the chamber). The EtOH was at 0 ${}^{\circ }$C, the calculated vapour pressure was 11.76 mmHg, while the temperature of the chamber remained at 27 ${}^{\circ }$C, in the darkness at atmospheric pressure. The results of the sensors N01 and P10 are shown in Figure 6a,b, the responses are labeled adding a letter, A or B, to distinguish if the vapor pressure of the EtOH was 58.73 or 11.76 mmHg, respectively. Both sensors had a less intense response than the previous, but they have a similar shape. A deeper study of the sensitivity correlated with the change on the thickness of the porous layer is necessary to determinate the limit and the behavior of the sensitivity. The sensitivity of the porous silicon (with a constant thickness of the porous layer) to organic vapours has been studied by different authors [27,28,29].

In order to reproduce the experimental response, we proposed the resistor–capacitor (RC) circuit shown in Figure 7a. The capacitor (*C*) was related to the electrode pattern and the separation distance; therefore, we assumed that it was a fixed value and did not change with the target gas presence. We proposed the use of a fixed resistance, named substrate resistivity ($R_{sb}$), whose value depends on the silicon resistivity (the crystalline silicon under the porous silicon plays the substrate role). A second resistance related to the sensor surface ($R_{sr}$) was proposed. The $R_{sr}$ value depended on the superficial reactions during the sensing due to the target gas interaction. Thereby, $R_{sb}$ and $R_{sr}$ can be modeled as parallel resistors $R=R_{sb}\left|\right|R_{sr}$. First, the influence of $R_{sr}$ (redox reaction) vs. a fixed value of $R_{sb}$ (substrate resistivity) is shown in Figure 7b. We observed that if $R_{sb}$ was low (compared with $R_{sr}$), the change on the current due to the redox reactions had no effect on the total resistance and the change of the electrical response was negligible.

The RC charge (Equation (Equation 2)) and discharge (Equation (Equation 3)) circuit responses were simulated and normalized. The variations of *C* or *R*, taking one of them as a constant, are shown in Figure 8a,b, respectively. The simulated response was obtained by t variations from t = 0 to t = 600 s (10 min), using first Equation (Equation 2), and then Equation (Equation 3). These equations simulate the experimental exposure and recovery periods, where each period is of 10 min. A cycle of 20 min was formed with the exposure and recovery periods. Three simulated cycles, shown in Figure 8b, reproduced the experimental response, where the time of the cleaning flux was not enough to return the response to the initial value due to a remnant charge in *C*. In the experiment, this behavior was related with residual physisorbed molecules on the sensor as a shift response. For the next cycles, *C* has an initial condition imposed by the remnant charge. In the experiment, the charge carrier concentration changed when the target gas interacted with the sensor surface and the redox reaction occurs. A sensible simulated response can be obtained doing $R_{sr}=R_{sb}$ if possible. The velocity of the response (charge and discharge) was imposed by the RC value. In our experiments, the response intensity depended on the thickness of the porous layer and the silicon conductivity type.
(2)$$R_{exp}=I_{max}(1-\exp\left(\frac{-t}{RC}\right))$$
(3)$$R_{rec}=I_{max}(\exp\left(\frac{-t}{RC}\right))$$

## 4. Conclusions

For the same target gas (EtOH), the intensity of the response and the stability depended on the thickness of the porous layer, and the silicon conductivity type. Our experiments showed that if the semiconductor material of the sensor was n-type and the target gas was a reducing gas, the response intensity increased as the thickness of the porous layer increased. However, if the semiconductor material of the sensor was p-type and the target gas was a reducing gas, the response intensity, the dead time, and the recovery were inversely proportional to the thickness of the porous layer. These results can be used to select a semiconductor material to fabricate a sensor based on the type of gas (oxidizing or reducing) to be sensed. In the case of a p-type semiconductor material to sense a reducing gas, a reduced thickness of the porous layer (specific area) must be used to obtain a more intense response with a shorter dead time and better recovery. 

## Figures and Tables

**Figure 1 sensors-20-04942-f001:**
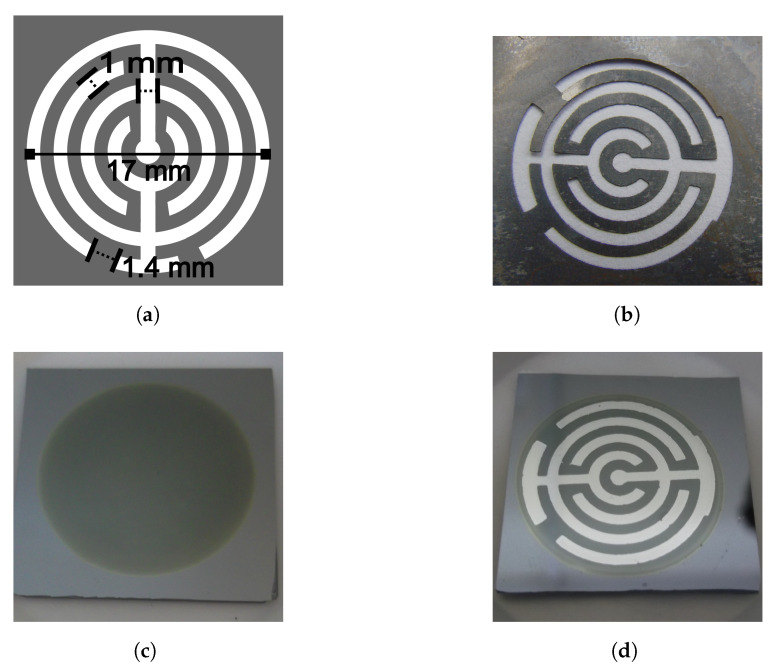
Metallic electrodes. (**a**) The geometric pattern consisted of two electrodes, each electrode was formed with two concentric no-closed rings of 1 mm width, a line of the same width connected them. (**b**) A photograph of the mask used to deposit the electrodes, made by lithography. (**c**) A photograph of the top view of a porous silicon sample; the area of the circle is porous silicon. The bulk silicon around and under the porous silicon serves as the substrate. (**d**) A photograph of a porous sensor with aluminium electrodes deposited on the porous area. The diameter of the outer no-closed ring was 17 mm.

**Figure 2 sensors-20-04942-f002:**
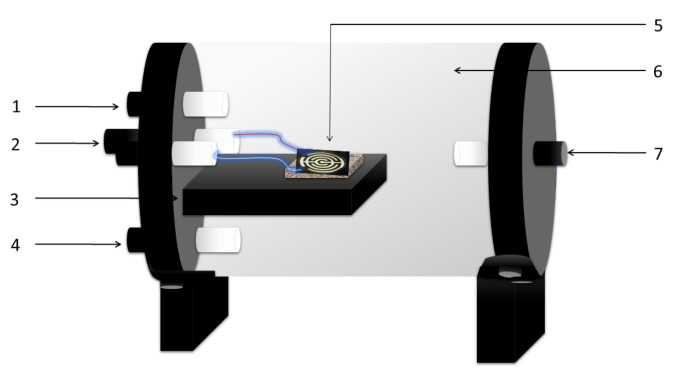
Sensing reactor scheme. 1. Gas input, 2. Electric connections, 3. Sensor holder, 4. Temperature measurement, 5. Sensor, 6. Sealed reactor (quartz), 7. Gas exhaust. The characterizations were made in darkness at 27 ${}^{\circ }$C.

**Figure 3 sensors-20-04942-f003:**
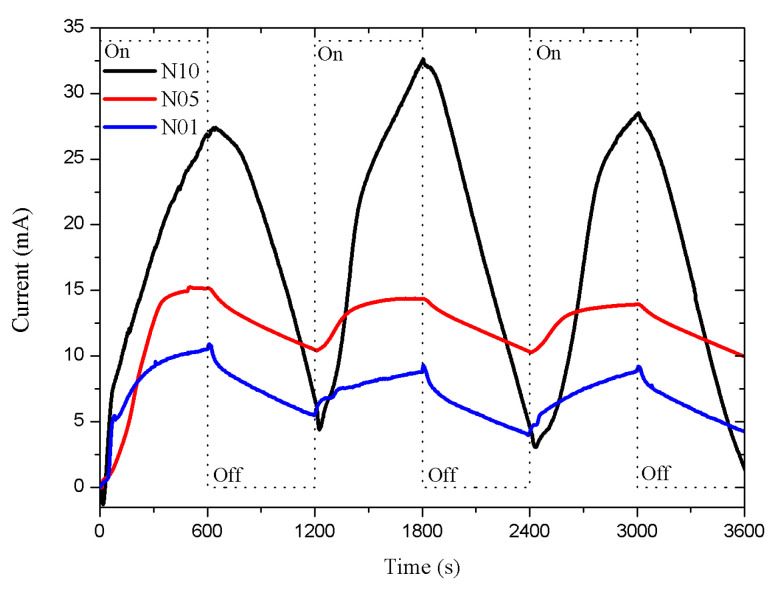
Porous sensor response, Si n-type with 0.0015–0.004 Ω·cm resistivity, connected to a 5 V DC supply. The porosity was 45% and the thicknesses of the porous layers were 1, 5, and 10 μm; labeled as N01, N05, and N10, respectively. The dead time was around 30 s.

**Figure 4 sensors-20-04942-f004:**
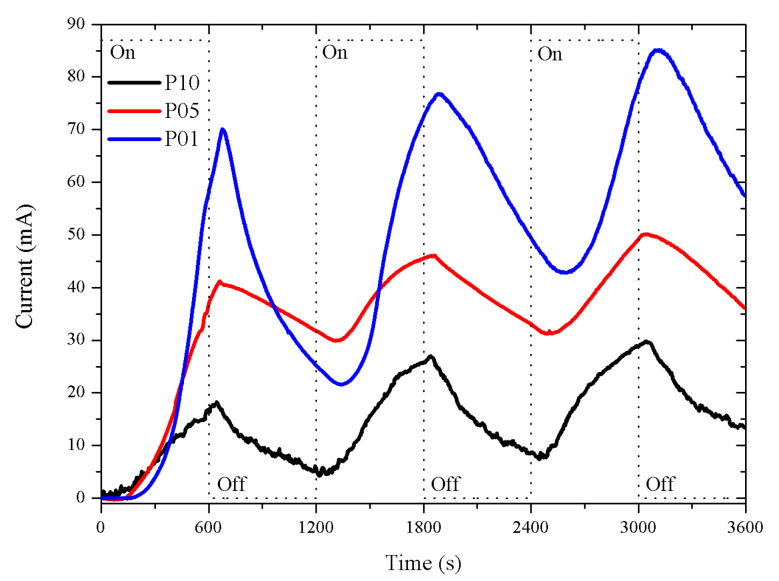
Porous sensor response, Si p-type with 0.01–0.02 Ω·cm resistivity, connected to a 5 V DC supply. The porosity was 45% and the thicknesses of the porous layers were 1, 5, and 10 μm; labeled as P01, P05, and P10, with dead times of 115, 152, and 247 s, respectively.

**Figure 5 sensors-20-04942-f005:**
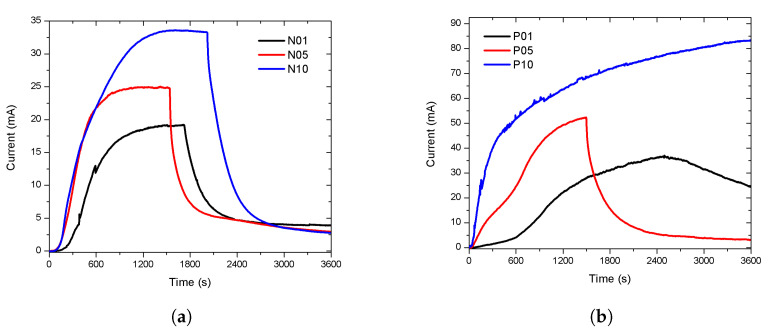
The stability of the response. The sensors were exposed to EtOH vapour for an extended duration (more than two hours) arbitrarily. (**a**) The response of the n-type porous sensors.The responses reached stability and they recovered when the flow of EtOH stops. (**b**) The response of the p-type porous. The stability was not achieved. The exposure was stopped after 2, 4, and 6 h for P05, P01, and P10, respectively. Only the recovery of P05 was observed, due to the long time of the measurements.

**Figure 6 sensors-20-04942-f006:**
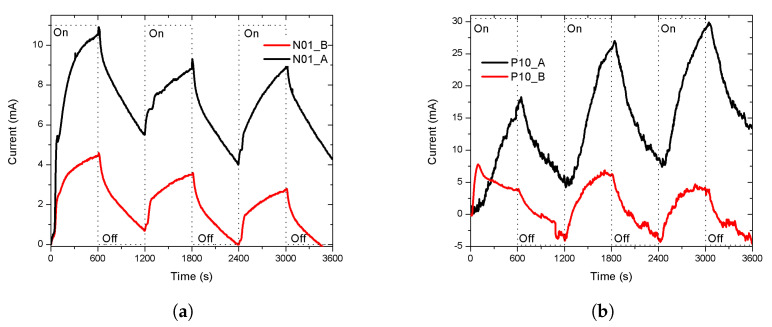
The sensitivity of the response at a lower vapour pressure (11.76 mmHg). The temperature of the chamber remained at 27 °C (**a**) Porous sensor N01. The response of the sensor at 58.73 mmHg, labeled as N01_A, had a similar shape than its response at 11.76 mmHg, labeled as N01_B, but was less intense. (**b**) Porous sensor P01. The first cycle (first 20 min) of the response of the sensor at 58.73 mmHg, labeled as P10_A, was different than its response at 11.76 mmHg, labeled as P10_B, and less intense. The current, in the cycles, was under 0 mA because the baseline was subtracted from the original response so that the intensity of the responses started at zero and can be compared.

**Figure 7 sensors-20-04942-f007:**
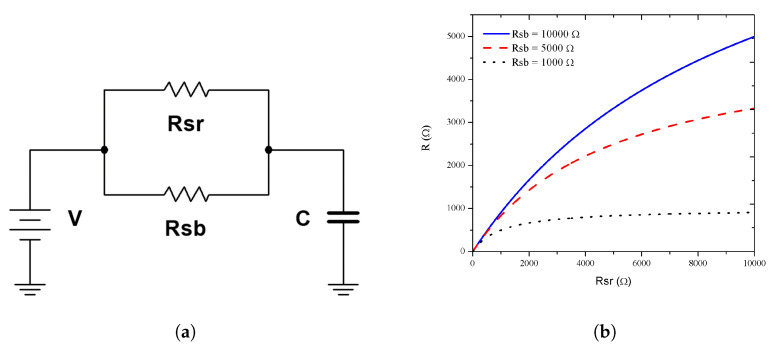
Electrical model. (**a**) The *RC* circuit used to obtain the experimental response where *R* was the parallel of *R_sr_* with *R_sb_*. (**b**) *R* versus *R_sr_* with fixed *R_sb_* of 1, 5, and 10 kΩ.

**Figure 8 sensors-20-04942-f008:**
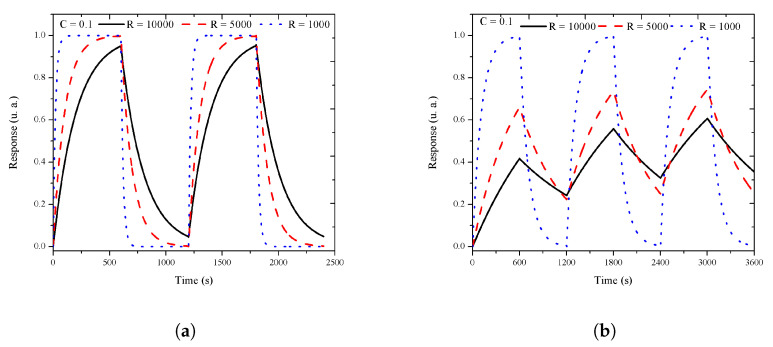
The *RC* circuit response. (**a**) Normalized response R vs. time with *C* = 0.1 F and R = 1, 5, and 10 kΩ. (**b**) Three cycles of 20 min, with 10 min for the exposure period and 10 for the recovery period.

**Table 1 sensors-20-04942-t001:** Porous silicon samples.

Type	Sample	Thickness (μm)	Current Density (mA/cm${}^{2}$)	Time (s)	Porosity (%)
	N01	1	10	82	45
n	N05	5	10	413	45
	N10	10	10	825	45
	P01	1	13.6	64	45
p	P05	5	13.6	321	45
	P10	10	13.6	642	45

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
