# Peer review of "Porous Silicon Gas Sensors: The Role of the Layer Thickness and the Silicon Conductivity"

_sensors, 2020, doi:10.3390/s20174942_

Round 1

Reviewer 1 Report

This paper present the experimental test of a resistive gas sensor based on a porous silicon “sensitive” layer. This work which presents the sensor as well as a simple electrical modelization is interesting because of the use of the porous silicon wafer which induces a “high” specific area which may improve the sensitivity of the sensor.

Nevertheless, even if the results and the setup of the experiments are well described, my general comment about this paper is that the most interesting part (i.e. the Porous Silicon - PS) is not well expanded. A specific section on the characterization of each constructed porous layers could help to appreciate this “sensing layer”. The author indicates that the porosity is about 45% with three thicknesses between 1 and 10µm. Is the experimental process of PS fabrication can conclude about these parameters, physical and morphological characterization of the PS can enhance the quality of the paper.

About the sensor itself, it’s difficult to appreciate the final device. If the description in the text is comprehensible, the pictures and the draw in Fig. 1 is not clear. First it is difficult to understand the caption (dimensions are in Fig. 1a or Fig. 1b?). Moreover a schematic section drawing each layer of the sensor (substrate?, PS, metallization) with the thicknesses could help to appreciate the realized device. Finally it is difficult to appreciate the dimension of the sensor in the picture of Fig.1b. The maximum diameter of the electrode is 17 mm, I think it misses a reference close to the whole sensor in the picture of Fig. 1 b (why not a coin, or a finger, or a part of a ruler).

About the experiments: the setup is well described, but it misses a “control experiment” without ethanol, but in the same temperature, pressure, and humidity conditions (as insofar as possible). Indeed, the ethanol is stored in a thermostatic bath before injection (bubbling). What is the temperature of the bath? What is the temperature in the chamber of analysis during each experiment (especially if the chamber of analysis has a temperature sensor)? Is the author can estimate that the influence of physical parameters (temperature pressure, humidity) have no (or a marginal) modification of the response of the sensor (especially when the flow rate is not the same between nitrogen and ethanol, moreover when the ethanol have to be bubbled?)?

I suggest the author do no use acronym or expression before or without the definition (ex RCA at line 41, or dead time at line 91, substrate, etc…). The ref4 at line 20 seems not appropriate, please double-check each reference.

I suggest that the authors add additional experiments and information before the publication of this interesting paper.

Author Response

First, we want to thank the reviewer for its time and valuable comments.

Some changes were made considering the comments of the reviewers, these were highlighted in the manuscript.

Please, see the attachment were we response to the points, and the changes in the manuscript.

Reviewer 2 Report

This paper on using porous silicon sensors is interesting. However, there are a few clarifications I would like to see before it is published. I have listed them below in no particular order:

- A key parameters (as the authors themselves have stated) is the surface area of the porous silicon. I was expecting to see this reported in the manuscript but was dissappointed not to find it. It is important to report details about the porous silicon so that they can be comparable to other porous silicon (or even other porous materials used for sensing). Please characterize and report the surface area (using the BET method or another suitable method) and also report the pore size distribution.

- The authors talked about the beneficial stability of porous silicon, relative to metal-oxide sensors... but there is no information in the manuscript about the specifics of their stability. Have any stability tests on porous silicon been done? How stable are these materials for sensing applications? How many cycles can we expect before their behavior starts changing (and for what adsorbates!).

- In general, there is a lack of characteriziation of the porous material itself. I would like to see scanning electron microscope images of the porous silicon (or high magnification optical microscope images if they are able to visualize the pores). Any other materials characterization that can shed light on the surface structure or pore morphology of this porous silicon would strengthen the paper.

- In section 2.2 is "9.9999%" a typo? I assume the authors meant "99.9999%"

- I did not understand the metallization procedure. What parts were covered in aluminum and what parts were not? A schematic figure showing this process clearly would be helpful. Is aluminum being deposited onto the porous silicon?

I look forward to seeing this paper published after it is revised.

Author Response

(The authors gave the same response as above.)

Reviewer 3 Report

The topic that “Porous Silicon Gas Sensors: The Role of the Layer Thickness and the Silicon Conductivity” is valuable. However, the research carried on in the article is insufficient to support this topic. There are so many points require further supplement and explanation. I do not think the manuscript is worth publishing on “Sensors” yet.

The comments are as below:

  1. The first area in need of improvement is related to the editorial aspect. It contains a number of typos and grammatical errors. Careful proof-reading is recommended.
  2. The view that “the specific area was varied changing the thickness of the porous layer, this means a variation of the length of the pores” is untenable. The authors should reconsider carefully the effect of the layer thickness for the sensing performances and fully explain it.
  3. The conclusion of line 109-118 is bland and irrelevant to the topic. The authors should reconsider the value of this part.
  4. What does Fig.6b mean? Please explain the meaning of this sentence “The simulated response show in Figure 6b was fitted with the exposure and recovery periods, which response was divided in 3 cycles of 20 minutes, 10 related with the exposure flux and 10 with the cleaning flux.”

Author Response

(The authors gave the same response as above.)

Round 2

Reviewer 1 Report

The authors added more precisions in the article which facilitate the understanding, and read of the paper. 

Author Response

Dear reviewer,
we thank you very much for your time.

The manuscript was submitted for English editing and an English editing certificate was obtained (attached). We checked the changes and ensured that the meanings were retained.

Best regards 

Francisco Ramírez 

Reviewer 2 Report

Looks great now, thanks for making the revisions.

Author Response

(The authors gave the same response as above.)

Reviewer 3 Report

Extensive editing of English language and style required

Author Response

(The authors gave the same response as above.)
